# Carriage of Multidrug Resistance Staphylococci in Shelter Dogs in Timisoara, Romania

**DOI:** 10.3390/antibiotics10070801

**Published:** 2021-07-01

**Authors:** Dégi János, Herman Viorel, Iancu Ionica, Pascu Corina, Florea Tiana, Dascălu Roxana

**Affiliations:** Faculty of Veterinary Medicine, Banat’s University of Agricultural Sciences and Veterinary Medicine Timișoara, Calea Aradului 119, 300645 Timișoara, Romania; viorelherman@usab-tm.ro (H.V.); iancu.ionica@usab-tm.ro (I.I.); corinapascu@usab-tm.ro (P.C.); roxana-dascalu@usab-tm.ro (D.R.)

**Keywords:** staphylococci, dogs, resistance, public health

## Abstract

The present study aimed to determine the prevalence of *Staphylococcus* species, which pose risks for public health, by evaluating skin samples collected from dogs in an animal shelter in Timisoara. Skin samples were taken from 78 dogs, which were either clinically healthy or suffering from dermatological conditions. *Staphylococcus* spp. was isolated and recognized based on conventional methods based on colony appearance, microscopic morphology, sugar fermentation, and coagulase activity. Following biochemical analysis, *Staphylococcus* isolates were subject to PCR tests to detect *sa-f* and *sa-r* genes to confirm the isolates to genus level. The typical colonies were identified to species level using biochemical methods, namely the VITEK^®^2 ID-GP64 identification card (bioMerieux, France). The phenotypic antimicrobial resistance profiling was performed using the VITEK^®^2 AST GP Gram-positive specific bacteria card (bioMerieux, France). Forty-three samples were confirmed as positive for *Staphylococcus* spp. *Staphylococcus* isolates were classified into the following categories: *S. aureus*, *S. pseudintermedius*, *S. intermedius*, *S. epidermitis*, *S. haemolyticus,* and *S. hyicus*. Eight (18.60%, 8/43) out of all the samples harbored the *mecA* gene, highlighting the distribution among isolated staphylococcal species: *Staphylococcus pseudintermedius* (4/43, 9.30%), *Staphylococcus intermedius* (1/43, 2.32%) and *Staphylococcus aureus* (3/43, 9.30%), respectively. The phenomenon of resistance was present, to the following antimicrobial agents: erythromycin (38/43, 88.37%), benzylpenicillin, kanamycin, and tetracycline with 37 strains (37/43, 86.04%), gentamycin (30/43, 69.76%), chloramphenicol (29/43, 67.44%), trimethoprim/sulfamethoxazole (27/43, 62.79%), ampicillin (26/43, 60,46%), rifampicin (25/43, 58,13%), imipenem (14/43, 32,55%), nitrofurantoin (11/43, 25.58%), oxacillin (8/43, 18.60%), vancomycin (4/43, 9.30%) and clindamycin (3/43, 6.97%), respectively. The presence of multidrug-resistant zoonotic staphylococci in clinically healthy dogs and dogs with skin lesions is an animal health and human health concern.

## 1. Introduction

Bacteria of the genus *Staphylococcus* are Gram-positive, facultative anaerobic cocci and are members of normal cutaneous and mucosal microbiota of mammals and birds. Various *Staphylococcus* species will colonize most animals with a certain site-predisposition [1,2].

Along with the outset of antimicrobial drug use in the practice of modern human and veterinary medicine, staphylococci have undergone evolutionary processes in response to the presence of antimicrobial drugs in biological systems. This evolution implied a novel development or acquisition of antimicrobial drug resistance mechanisms, in addition to the amplification and proliferation of clinically important strains of pathogenic staphylococci affecting human and animal populations.

Some degree of antimicrobial resistance has been documented within all *Staphylococcus* species that infect humans and domestic animals.

Methicillin-resistant *Staphylococcus aureus* (*S. aureus*) (MRSA) is a major healthcare-associated pathogen worldwide and has increased in incidence dramatically over the last decade [1,2]. Companion animals have been implicated more frequently as potential reservoirs of MRSA than other livestock [3,4]. A 0–4% prevalence rate of MRSA in dogs has been reported [5,6,7]. Other reports demonstrated MRSA at a higher prevalence (~9%) in pets and veterinary staff [8,9]. The nasal and skin carriage of MRSA plays a crucial role in the epidemiology and pathogenesis of community-associated infection.

In Romania, the antibiotic susceptibility testing of bacterial strains isolated from dogs (including stray dogs) was focused on the *Staphylococcus* genus. It was carried out mainly in veterinary teaching units during the past twenty years [4,5]. In addition, a recent review strengthens the imperative need for an integrated surveillance system in the One Health context of antimicrobial resistance profiles of zoonotic bacteria including *Staphylococcus* spp. [10]. Hence, the present study was designed to investigate the incidence of the staphylococcal microbiota in shelter dogs that were either healthy or presenting a skin condition and provide new information on their antimicrobial resistance profile (AMR).

## 2. Results

A total of 78 skin samples were investigated for the presence of *Staphylococcus* spp., between June and October 2019, and 43 (55.12%) of these samples were found to be positive for *Staphylococcus* spp.

Species identification was performed based on colony and microscopic morphology, sugar fermentation, pigment production in blood agar, and coagulase activities [6].

All the isolates (*n* = 43) were confirmed as *Staphylococcus* spp. by PCR amplification of the *invA* genes, which generated amplicons of 284 bp. A total of 43 samples were confirmed positive for *Staphylococcus* spp. by conventional and molecular methods. However, we were unable to amplify DNA of interest extracted from 19 skin sample isolates, which, following primary processing (inoculated on culture media: Columbia agar with 5% Sheep Blood, MacConkey agar and Chapman Mannitol Salt agar, respectively) proved positive for other bacterial species. Similarly, we could not use 16 other samples that tested negative.

Overall, 43 (43/78, 55.12%) isolates of *Staphylococcus* spp. were identified (Table 1).

The result obtained using the Vitek 2^®^ ID-GP card may be categorized into one of several confidence levels (excellent, very good, good, acceptable, good, low, unidentified, and error), specified in Table 2.

*S**. pseudintermedius*, *S**. intermedius,* and *S. aureus,* respectively, are the main species isolated from the skin of shelter dogs, with usefulness in monitoring the risk factors implied by staphylococcal infections. The *Staphylococcus pseudintermedius* was detected in 48.83% (21/43) of samples, *Staphylococcus intermedius* in 27.90% (12/43), *Staphylococcus aureus* in 11.62% (5/43), *Staphylococcus epidermidis* in 9.01% (3/43), one *Staphylococcus hyicus* isolate (2.32%, 1/43) and one isolate was *Staphylococcus haemolyticus* (2.32, 1/43) (Table 2).

PCR genotyping confirmed that the isolates were positive for *S. pseudintermedius* (21 isolates), *S. intermedius* (12 isolates), *S. aureus* (5 isolates), *S. epidermidis* (3 isolates), *S. hyicus* and *S.* *haemolyticus* (each with one isolate).

Detection of the *mecA* gene represents the standard procedure for determining resistance to methicillin. From the total number of isolates, 8 (18.60%, 8/43) harbored the mecA gene, highlighting the distribution of isolated staphylococcal species: Staphylococcus pseudintermedius (4/43, 9.30%), Staphylococcus intermedius (1/43, 2.32%) and Staphylococcus aureus (3/43, 9.30%).

The most common lesions, which also served as collection sites for *Staphylococcus* spp. strains were as follows: erythema, 25.58% (11/43 strains); pustules, 13.05% (6/43 strains); hyperkeratosis lesions 6.97% (3/43 strains) and alopecia 4.65% (2/43 strains), respectively (Table 3). Forty-six stray dogs, which had their skin sampled, showed skin lesions, while thirty-two of them did not have associated skin diseases.

All of the 43 studied *Staphylococcus* isolates were resistant to at least three antimicrobial agents. All identified strains were susceptible to ampicillin/sulbactam, enrofloxacin, marbofloxacin, and mupirocin. The phenomenon of resistance was manifested against several antibiotics such as: erythromycin (38/43, 88.37%), benzylpenicillin, kanamycin, and tetracycline in the case of 37 strains (37/43, 86.04%), gentamycin (30/43, 69.76%), chloramphenicol (29/43, 67.44%), trimethoprim/sulfamethoxazole (27/43, 62.79%), ampicillin (26/43, 60.46%), rifampicin (25/43, 58.13%), imipenem (14/43, 32.55%), nitrofurantoin (11/43, 25.58%), oxacillin (8/43, 18.60%), vancomycin (4/43, 9.30%), clindamycin and fusidic acid (3/43, 6.97%), respectively (Table 4).

Statistical analysis was performed using a Chi-square (χ2) test in SPSS version 21.0 for Windows (SPSS Inc., Chicago, IL, USA) in order to evaluate the probability that *S. pseudintermedius* and *S. intermedius* demonstrate resistance towards the tested antimicrobials. The differences with *p* < 0.05 were considered statistically significant.

Following genetic analysis, it was revealed that all eight isolates, phenotypically demonstrating oxacillin resistance, harbored the *mecA* gene.

Vancomycin is the drug of choice when it comes to treating invasive MRSA infections [11]. Out of the eight isolates of MRSA included in the study, none have developed resistance to vancomycin (VRSA) according to the revised CLSI guidelines. The MIC of vancomycin-resistant strains ranged from 0.5 µg/mL to 4 µg/mL.

## 3. Discussion

The nature of animal shelters that keep many animals in a confined space increases bacterial transmission. Proper preventive care remains a challenge to the resident animals and the people working and visiting there [12]. Another study conducted by Gonzales-Dominguez et al. [8] describes that 100% of dogs with superficial skin infections tested positive for *Staphylococcus* species, suggesting a high incidence of this genus in dogs dermatologic pathologies.

The recorded *Staphylococcus* isolates in the present study expressed resistance towards methicillin and other antimicrobials (e.g., gentamicin, erythromycin, and tetracycline) commonly used to treat skin infections. In conclusion, MRSA can be considered a frequently found microorganism in shelter animals, with widespread resistance to the currently used antibiotics. [13].

*Staphylococcus* spp., considered a ubiquitous bacterium that covers all ecological niches, might be accompanied by AMR characteristics. The complex relationships between bacterial species from different “environments” facilitate this genetic flow, extending AMR between humans, animals, and the environment, resulting in a general public health issue [14]. *Staphylococcus* methicillin resistance is one of these significant problems of society. An animal shelter may act as a reservoir of MRSA, considering some particular setting, with a high colonization rate, which can reach 7.8% in shelter dogs (reviewed by [15,16,17]). Accurate and timely determination of methicillin resistance is of crucial importance in the prognosis of *S. aureus* infections [18,19]. These strains have spread in the community (MRSA-CA-MRSA associated with the community) and have been incriminated in cases of severe infection in healthy individuals [20,21,22,23]. The rate of colonization with this bacterium in dogs varies significantly; however, about 10% of dogs can act as carriers of *S. aureus* [11]. The growing concern about the spread of MRSA in communities has led to the establishment of particular recommendations for surveillance, including research into the rate of colonization in healthy dogs (including stray dogs present in public areas, which may later be adopted).

Exposure to shelter dogs or the living environment of these dogs can be considered a risk factor for colonization with CA-MRSA strains; thus, it is essential to identify community and environmental reservoirs and sources. In this regard, it is imperative to precisely determine the phenotype of resistance and the mechanisms underlying resistance, which are essential in therapy and public health. The most widely used method, considered a gold standard in identifying MRSA strains, is to identify the *mecA* gene.

The most important mechanism of resistance in staphylococci is resistance to methicillin, which, clinically speaking, implies resistance to all β-lactam antibiotics, often accompanied by resistance to many other groups of antimicrobial agents. It is estimated that 30% of the world’s population could be colonized with *S*. *aureus* [11]. Rajwin Raja Kanagarajah et al. [12], in a study on antibiotic profiling of methicillin resistant *Staphylococcus*
*aureus* (MRSA) isolates in stray canines and felines from India, isolated 283 strains of staphylococci, of which 33 (11.66%) were MRSA, using CromMRSA agar medium (Chromagar, Oxoid, United Kingdom). Canine MRSA isolates exhibited resistance, in decreasing order, to methicillin (100%), ceftazidime (81.82%), enrofloxacin (78.79%), oxacillin (60.61%), and vancomycin (0%). Over the years, frequent and random use of vancomycin may bear the responsibility for the emergence of vancomycin resistance among the *S*. *aureus* isolates [11].

*Staphylococcus pseudintermedius* is an opportunistic pathogen that has been identified as an infectious agent or colonizer, mainly in dogs. *S. pseudintermedius* has also been detected in humans, more specifically in people that are in close contact with dogs [7]. Multidrug methicillin-resistant *Staphylococcus pseudintermedius* (MRSP) detection has rapidly increased among microbial specimens from pets across Europe. (reviewed by [24]). *S. pseudintermedius* carriage is frequently more significant than 80% in some healthy dog populations. It is also an opportunistic pathogen that causes severe and necrotizing infections and is commonly found in the skin, ears, bones, and post-surgical abscesses [25]. Considered as a risk for humans, there is a possibility for it to have been misidentified as *S. aureus* in human infections [25]. The dogs may be colonized or infected with MRSP on healthy skin, fur, and mucosae even after the infections have healed spreading MRSP to their environment, other species, and humans.

It also serves as a reservoir of re-infection with MRSP. Although MRSA transmission in humans is well understood, little is known about MRSP carriage and it’s spread in dogs (reviewed by [16,26]). In the recent literature, *S. pseudintermedius* is one of the important pathogens of zoonotic origin that causes wound and skin infections in humans. According to the literature, up to 90% of healthy dogs may be colonized with *S. pseudintermedius* [27]. Antimicrobial resistance is being caused by the unjustified use of antimicrobials in companion animals. *S. pseudintermedius* is another element in the same chain of evolving drug resistance. It is multidrug-resistant, capable of transmitting from animals to humans, and possesses many *S. aureus* virulence factors [22]. Although data on drug resistance and pathogenesis of *S. pseudintermedius* are not sufficient, it is imperative to identify the pathogen correctly. Resistance to beta-lactam antibiotics (including methicillin) is a significant concern. Methicillin resistance is a significant concern in developed countries when antibiotic usage is unregulated. Multidrug tolerant isolates from clinical specimens have been published in European studies [28,29,30].

Resistance to erythromycin, clindamycin, tetracycline, trimethoprim-sulfamethoxazole, and ciprofloxacin was identified among *S. pseudintermedius* strains by Lozano et al. [7] as well. The matter of drug resistance among strains of *Staphylococcus* is an increasingly serious concern. However, the susceptibility of *Staphylococcus* to antimicrobial drugs varies among countries and regions [8,9,10]. The recently introduced term ‘One Health’ states that only ‘One’ health is shared by humans, animals, and the ecosystem, explicitly insinuating that anything that affects one will ultimately affect all three components [31,32,33]. Currently, the One Health concept is advocated for antimicrobial resistance [31,32]. Therefore, One Health-based approach to fight antimicrobial resistance has been promoted as a global challenge [33]. To support the application of the One Health-based approach to combat antimicrobial resistance, research on the human-animal–environmental interface is essential [33]. There at present are no shelter-specific MRSA and MRSP control guiding principles but existing Guidelines for Standards of Care in Animal Shelters, adopted by The Associations of Shelters Veterinarians. However, general infection control strategies should always be used to prevent infectious agent spread among shelter animals and decrease the probability of environmental contamination. Such should aid in the control of MRSA and MRSP if the bacteria are present in the animals or environment and will help to protect both animal and human populations in the shelter environment [34].

The detection of MRSA in a 9.30% percentage in our study indicates a high prevalence of MRSA colonization rate in the shelter dogs population tested, compared to others study, findings of 1–2% MRSA prevalence in hospitalized dogs [35], 0–0.4% in healthy animals [36], respectively 7.8% in shelter dogs (reviewed by [15,16,17]). MRSP prevalence was between 0.5–16.7% in healthy dogs [37]. The MRSP prevalence in the shelter dogs of our study (9.30%) is within the range of most studies. Animals from shelters may be more exposed to the risk of colonization or even of developing infections caused by various agents (e.g., zoonotic *Staphylococcus* spp.). Several reasons may stand behind this exposure: high density of individuals, the possibility of nosocomial transmission from shelter staff members or volunteers, suboptimal cleaning and disinfecting protocols, and the unidentified carrier status of high numbers of animals, or a stressful environment [38].

## 4. Conclusions

This study shows that sheltered animals are likely an unusual source of infections for shelter staff or potential adopters. Methicillin-resistant staphylococci are a substantial public health risk. Animals in a shelter may, in the end, end up in the homes of a wide range of possible adopters: homes with young children, healthy humans, and immunocompromised, respectively. As well, other species of coagulase-positive staphylococci have been isolated, such as *S. intermedius* and *S. pseudintermedius*, considered zoonotic, producing various infections in both dogs and humans.

The data from this study reveal that resistance to beta-lactams and other classes of antibiotics are omnipresent in the medically significant species of pathogenic staphylococci that colonize shelter dogs. This situation poses a high risk of transmission of MDR strains or mobile elements encoding resistance in humans, being in close contact with those animals and their microbiota. The results highlight the need for reliable implementation of the One Health concept; considering all the interconnections between human populations and shelter dogs, these pathogenic staphylococcal species and their resistance mechanisms can be transmitted.

The study results are confined to the individual, single shelter, and future research is essential to establish whether rates similar across shelters, regions, and other source animals, such as pet shops, breeders, were similar. We hope that this study will increase public health-related responsiveness regarding biosecurity measures and manure or animal shelter waste management.

## 5. Materials and Methods

### 5.1. Sample Collection

The dogs that made the subject of this study were in the custody of two of the community dog collection centers, under the management of Timisoara City Hall, in collaboration with the University Veterinary Clinics of the Timisoara Faculty of the Timisoara Faculty of Veterinary Medicine.

All dogs enrolled in the study were taken from the streets of Timisoara and housed in two specialized centers for stray dogs.

The samples were meant to serve as material for evaluating the possible public health risk posed by these animals as well as for establishing the rate of zoonotic staphylococcal carriage among the population of stray dogs. All methods were performed in accordance with the relevant guidelines and regulations. A dog with skin lesions was defined as an animal currently presenting a skin condition: erythema, scaling, alopecia, pruritus, crusts, pustules, and hyperkeratosis, while dogs without skin lesions were defined as healthy. A total of 78 skin samples were collected from shelter dogs between June and October 2019, and screened in order to obtain an estimative prevalence of zoonotic drug-resistant *Staphylococcus* spp., in an urban area of Timisoara City (DMS Coordinate: 45°45’13.39” N, 21°13’32.56” E), western Romania. The samples from animals that appeared clinically healthy (stray dogs without skin lesions) were collected with the help of students that were enrolled in the practice program at the University Veterinary Clinic from the Faculty of Veterinary Medicine, Timisoara. The sample-collection protocol was chosen in agreement with regulations imposed by the Romanian Veterinary College (protocol numbers 34/1.12.2012) and according to current practice at the University Veterinary Clinics of the Faculty of Veterinary Medicine from Timisoara.

A standard procedure was established to recruit both animals with and without skin lesions for this study.

The skin samples were obtained from dogs with skin lesions, following specialty evaluation, performed on-site at the dog shelter. The students were volunteers enrolled in the study after receiving proper training on collecting samples in compliance with all biosecurity measures [37].

### 5.2. Bacterial Isolation

*Staphylococcus* spp. was isolated using conventional methods [39], as mentioned in the protocols recommended by the samples, collected using the eSwab™ (Copan, Italy) transport systems, were subsequently stored in cooling containers and transported, according to the biological sample collection and transport guidelines, to the research laboratory of transmissible diseases in pets (B.6.d), where the samples were processed in the shortest possible time (max. 3 h post-collection).

The samples were processed in the Bacterial Diseases diagnostic laboratory (B.6.a), within the Department of Infectious Diseases and Preventive Medicine, of the Faculty of Veterinary Medicine Timisoara.

The samples were inoculated onto BD Columbia Agar plates, with 5% Sheep Blood (Becton Dickinson GmbH, Heidelberg, Germany), and incubated at 37 °C for 24 h, under aerobic conditions [39,40] The identification of staphylococci in primary culture, was based on colony morphology, appearance, type of hemolysis, and Gram staining.

The specific colonies that resulted from Columbia agar with 5% Sheep Blood were also inoculated on MacConkey agar (Thermo Fisher Scientific, UK) using a bacteriological loop and incubated at 35 °C, in an aerobic atmosphere, for 24 h. In addition, the production of “free coagulase” and the presence of the “clumping factor” was also determined using rapid slide agglutination tests such as Bactident Coagulase (Merck, Darmstadt, Germany) and Staphytect Plus (Oxoid, Ltd., Basingstoke, UK), respectively.

Following this preliminary stage, the colonies presenting phenotypical characteristics specific for *Staphylococcus* spp. were transferred on Chapman (mannitol salt) agar medium (Thermo Fisher Scientific, Basingstoke, UK) to facilitate the identification of pathogenic *Staphylococcus* strains based on their biochemical characteristics (mannitol fermentation) and for their purification. The plates were incubated at 37 °C, in an aerobic atmosphere, for 24 h.

In order to identify *Staphylococcus* species, we inoculated Vitek 2^®^ ID-GP identification cards (bioMérieux. Marcy l’Etoile, France) with *Staphylococcus* strains, according to the manufacturer’s guidelines and the results were analyzed and interpreted using the VT2- Software program, version R02. 03.. The Vitek 2^®^ ID-GP card is a 64-well card designed for the automated identification of most veterinary, clinically significant Gram-positive bacteria [41].

All characterized isolates have shown very good (%ID ≥ 99.0, T index ≥ 0.5) confidence levels.

### 5.3. Molecular Analyses

DNA was extracted from the biochemically identified isolates as *Staphylococcus* spp., using the PureLink™ Genomic DNA Mini Kit (Thermo Fisher Scientific, UK).

DNA was extracted from samples cultivated in Remel™ BHI Broth (Brain Hearth Infusion broth—Thermo Fisher Scientific, Basingstoke, UK) by inoculating presumed *Staphylococcus* spp. colonies, after incubation at 37 °C under conditions of aerobiosis, for 24 h [42].

#### Extraction of Template DNA

One milliliter of bacteria grown in 10 milliliters of Remel™ BHI Broth (Brain Hearth Infusion broth—Thermo Fisher Scientific, UK), at 37 °C in an aerobic atmosphere for 24 h, was dispensed aseptically in an Eppendorf tube. Bacterial genomic DNA was extracted using the Pure Link™ Genomic Lysis/Binding Buffer (Thermo Fisher Scientific, UK) boiling method, with a freshly-prepared proteinase K solution (10 mg/mL) as described previously by Rantakokko-Jalava and Jalava [31]. The DNA quantity and quality were determined using a Nano Drop ND-1000 spectrophotometer (Nano Drop^®^ Technologies, Thermo Fisher Scientific, UK), by measuring the absorbance at 260 nm [43].

PCR was done using genus specific primers, *sa-f* and *sa-r* genes of *Staphylococcus* 16S-1: 5′-GTGCCAGCAGCCGCGGTAA-3′ and 5′-AGACCCGGGAACGTATTCAC-3′. The primers for the *nucA* nuclease gene used in our study were *nuc-1*: 5′-TCAGCAAATGCATCACAAACAG-3′ and *nuc-2*: 5′-CGTAAATGCACTTGCTTCAGG-3′, to highlight the species *Staphylococcus aureus*. To highlight the *mecA* gene (gene that confers resistance to methicillin), we used primers: mecA-1: 5′-GGGATCATAGCGTCATTATTC-3′, respectively *mecA-2*: 5′-AACGATTGTGACACGATAGCC-3′ [44].

The preparation of the lysosomal digestion buffer was performed by using 200 µL of Pure Link™ Genomic Digestion Buffer (Thermo Fisher Scientific, Basingstoke, UK)/sample, adding fresh lysozyme to obtain a final lysozyme concentration of 20 mg/mL. Up to 2 × 10 Gram-negative cells were collected after centrifugation. The cell pellets were re-suspended in 180 µL Pure Link™ Genomic Digestion Buffer and mixed well, then incubated at 37 °C for half an hour. Twenty µL of proteinase K were then added and mixed thoroughly. Next, 200 µL Pure Link™ Genomic Lysis/Binding Buffer (Thermo Fisher Scientific, Basingstoke, UK) and mix, then incubate at 55 °C for 30 min. Next, 200 µL of ethanol of 96–100% concentration was added in the lysate content and mixed thoroughly for 5 s.

The enhanced PCR conditions consisted of an initial denaturation at 95 °C for 5 min followed by 32 cycles of denaturation at 95 °C for 1 min, annealing at 55 °C for 1 min, extension at 72 °C for 1 min, and a final extension at 72 °C for 10 min, using the My Cycler (Bio-Rad^®^, Dubai, United Arab Emirates) thermo cycler [45].

Amplicon control was performed by horizontal electrophoresis in the submerged system of 1.5% agarose gel electrophoresis at 120 V and 90 mA, over 60 min [46,47]. The amplified products were resolved by electrophoresis on 2.5% agarose gel, stained with ethidium bromide, and visualized under UV light using a gel documentation system (UV transilluminator—2035-2, Bio Olympics USA).

The strain *Staphylococcus aureus* ATCC^®^23235™, was used for the positive control (American Type Culture Collection, USA). The control used for the amplification of the *mecA* gene in our study was the *S. aureus* ATCC 49476 (*mecA* positive), and the β-lactamase positive *S. aureus* ATCC 29213 strain was used as a negative control. Sterile deionized water was the negative control for the PCR reactions.

### 5.4. Antimicrobials Susceptibility Test

The Vitek 2^®^, AST—GP69 Gram positive specific bacteria card (bioMérieux. Marcy l’Etoile, France), was used to determine antibiotic sensitivities for *Staphylococcus* isolates collected from shelter dogs, with European Union (EU) drug configuration for companion animals [48].

A total of 19 antimicrobial substances (minimum inhibitory concentration [MIC] from 13 classes were included in the study accordingly: ß lactams—benzylpenicillin (PCG; 0.03–0.5 µg/mL), oxacillin (OXA; 0.25–4 µg/mL), imipenem (IPM; 1–8 µg/mL), ampicillin (AM; 2–64 µg/mL), ampicillin/sulbactam (SAM; 2–64 µg/mL); aminoglycosides—gentamicin (GM; 0.5–16 µg/mL), kanamycin (K; 0.25–64 µg/mL); quinolones—enrofloxacin (ENR; 0.25–16 µg/mL), marbofloxacin (MBX; 0.25–8 µg/mL); steroids—fusidic acid (FUS; 1–16 µg/mL); glycopeptides—vancomycin (VAN; 0.25–8 µg/mL); macrolides—erythromycin (ERY; 0.25–16 µg/mL µg/mL), rifamycins—rifampicin (RIF; 0.5–8 µg/mL); lincomycins—clindamycin (CLI; 0.25–16 µg/mL), tetracyclines—tetracycline (TE; 2–32 µg/mL); sulfonamides—trimethoprim/sulfamethoxazole (SXT; 20–76 µg/mL); nitrofuran derivate—nitrofurantoin (FT; 16–512 µg/mL); pseudomonic acid derivatives—mupirocin (MUP; 0.06–512 µg/mL) and amphenicols—chloramphenicol (CHL; 4–32 µg/mL). The MIC at which an isolate is considered susceptible according to the Clinical Laboratory Standards Institute (CLSI) guidelines, CLSI M31-A4 2013 [49], based on the description of Humphries et al. [50].

Quality control was performed following the guidelines specified by the CLSI (CLSI, 2008) using *Staphylococcus aureus* ATCC^®^23235™. All susceptibility results obtained from quality control strains were within the quality control ranges.

This card was used with the Vitek 2^®^ Systems in clinical laboratories as an in vitro test to determine clinically significant aerobic Gram-positive bacteria’s susceptibility in character with the product information manual.

Isolates resistance to three or more classes of antimicrobials was classified as multidrug-resistant [51].

## Figures and Tables

**Table 1 antibiotics-10-00801-t001:** Distribution of positive skin samples for *Staphylococcus* spp., according to gender.

Gender	No. of Positive Samples/No. of Investigated (%)
Female	27/49 (55.10)
Male	16/29 (55.17)
Total	43/78 (55.12)

**Table 2 antibiotics-10-00801-t002:** Result of *Staphylococcus* spp. strains testing with Vitek 2^®^ ID-GP identification card (bioMérieux. Marcy l’Etoile, France).

Identified Strains	No. (%) of Strains Identified According to the Vitek 2^®^ ID-GP Indicators							
	N (%)	Excellent	Verry Good	Acceptable	Good	Low	Unidentified	Error
*Staphylococcus pseudintermedius*	21(48.83%)	18	3	-	-	-	-	-
*Staphylococcus intermedius*	12 (27.90%)	8	4					
*Staphylococcus aureus*	5 (11.62%)	3	2					
*Staphylococcus epidermidis*	3 (9.01%)	3	-					
*Staphylococcus haemolyticus*	1 (2.32%)	1	-					
*Staphylococcus hyicus*	1 (2.32%)	1						
Total	43 (100%)	34	9	-	-	-	-	-

**Table 3 antibiotics-10-00801-t003:** Distribution of skin samples that were positive for *Staphylococcus* spp., according to clinical aspects.

Clinical Result	Samples from Stray Dogs with Skin Lesions							Samples from Stray Dogs without Skin Lesions
	Erythema	Peeling	Alopecia	Pruritus	Scabs	Pustules	Hyperkeratosis	Without Clinical Signs
	21 (26.92%)	2 (2.56%)	9 (11.53%)	3 (3.84%)	2 (2.56%)	6 (7.69%)	3 (3.84%)	32 (41.02%)
Total collected samples	46 (58.97%)							32 (41.02%)
	78 (100%)							
Positive *Staphylococcus* spp. samples	11 (25.58%)	1 (2.32%)	2 (4.65%)	1 (2.32%)	1 (2.32%)	6 (13.05%)	3 (6.97%)	18 (41.87%)
Total samples	25 (58.13%)							18 (41.87%)
	43 (100%)							

**Table 4 antibiotics-10-00801-t004:** Patterns of resistance in *Staphylococcus* spp. (*n* = 43), obtained from stray dogs with skin lesions, exhibiting resistance for more than three antimicrobials.

No.	*Staphylococcus* spp.	No. of Isolates	Resistance to Antimicrobial Profile	No. of Antimicrobials Resistant
1.	*Staphylococcus pseudintermedius*	6	AM, PCG, CHL, ERY, K, GM, RIF *, TE, STX	9
2.	*Staphylococcus pseudintermedius*	4	AM, PCG, CHL, ERY, CLI, K, GM, OXA, TE, STX	10
3.	*Staphylococcus pseudintermedius*	3	AM, PCG, CHL, ERY, K, GM, TE, VAN *, FT, STX	10
4.	*Staphylococcus pseudintermedius*	3	AM, PCG, IPM, ERY, K, RIF *, TE, FUS,	8
5.	*Staphylococcus pseudintermedius*	3	PCG, CHL, ERY, K, RIF *, TE, IPM *	7
6.	*Staphylococcus pseudintermedius*	2	PCG, CHL, ERY, K, GM, RIF *, TE, FT	8
7.	*Staphylococcus intermedius*	4	AM, PCG, CHL, ERY, K, GM, TE, STX	8
8.	*Staphylococcus intermedius*	4	PCG, ERY, K, GM, RIF *, TE, STX	7
9.	*Staphylococcus intermedius*	3	AM, IPM *, GM, RIF, FT, VAN *	6
10.	*Staphylococcus intermedius*	1	PCG, CHL, CLI, ERY, K, OXA, STX	7
11.	*Staphylococcus aureus*	3	PCG, CHL, ERY, K, OXA, GM, TE, STX	8
12.	*Staphylococcus aureus*	1	AM, CHL, K, TE, STX, VAN *	6
13.	*Staphylococcus aureus*	1	PCG, CLI, GM, IPM *, FT, RIF *	6
14.	*Staphylococcus epidermidis*	2	PCG, CHL, IPM *, ERY, K, RIF *, TE	7
15.	*Staphylococcus epidermidis*	1	AM, PCG, FT, ERY, K, IPM *, CLI, TE	8
16.	*Staphylococcus haemolyticus*	1	AM, PCG, IPM *, ERY, FT, STX	6
17.	*Staphylococcus hyicus*	1	PCG, ERY, K, RIF *, TE	5

Legend: Ampicillin (AM), Ampicillin/Sulbactam (SAM), Benzylpenicillin (PCG), Clindamycin (CLI), Chloramphenicol (CHL), Erythromycin (ERY), Enrofloxacin (ENR), Fusidic acid (FUS), Gentamicin (GM), Imipenem (IPM), Kanamycin (K), marbofloxacin (MBX), Mupirocin (MUP), nitrofurantoin (FT), Oxacillin (OXA), Rifampicin (RIF), Tetracycline (TE), Vancomycin (VAN) and Trimethoprim/Sulfamethoxazole (SXT). * the first-line antimicrobial agents used in the treatment of MRSA.

## Data Availability

The datasets generated and analyzed during the current study are included within the article.

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
