# Peer review of "Carriage of Multidrug Resistance Staphylococci in Shelter Dogs in Timisoara, Romania"

_antibiotics, 2021, doi:10.3390/antibiotics10070801_

Round 1

Reviewer 1 Report

“Stray dogs, carriage of multi drug resistance of staphylococci and public health” by Janos et al presents a study analyzing the frequency of staph infection in an animal shelter animals.  Although it is an interesting paper from a veterinary perspective, it does not support the authors conclusion that this is an “extremely alarming matter”.  There is a lot missing from this paper, including statistical analysis and discussion about human-animal interaction. If this paper is accepted for publication, there are several items that should be addressed to improve its validity and readability.

Abstract and introduction

 Data on prevalence in animals and people is reported but there needs to be more information on why this is important to public health.  Could this result in death?  Is it spread among vet staff?  Zoonotic?  Lack of other antibiotics?  It doesn’t really generate interest or demonstrate what the importance is in this problem.

Line 10.  Please rephrase this sentence.

Line 40:  Would suggest including modern human and veterinary medicine

Line 47: Are agriculture animals included in this definition of domestic animals?

Line 49. Please provide a reference

60-63  Please reword sentence for clarity

Results

Line 73.  Sa-f and sa-r were mentioned in the abstract but where are they included again in the manuscript?

Lines 76-79. Please reword for clarity. 

Line 102-103 should be the first sentence in that paragraph, not the last

Line 113.  Why are stray dogs more important in environment?  I would have thought owned dogs would be more important given their proximity to humans in the households.

Lines 115-116 How much of an increase over how long?

Line 119.  Please give statistical significance values

Line 121  “induced complete sensitivity in all isolated strains”  Does this mean that the Staph spp strains were completely susceptible to these antibiotics?

Line 128.  Please provide a statistical significance between the antibiotics.

Discussion

The discussion almost seems like bullet points from a poster and doesn’t flow well.  A lot of the discussion is focused on S aureus.  Not sure why since that was not the most common Staph found in this study

  1. Beyond stray dog shelters, what other types of settings are considered “considerable reservoirs”? Dog racing kennels? 

Line 159.  Where  (“various skin regions”)on non stray dogs was Staph spp most commonly isolated?  Axila? Groin?  Where were most of the lesions on stray dogs from which Staph was isolated? 

Could you test the humans working in the dog shelter for staph and MRSA?

Lines 156-159.  Please include a brief description of what the previously published studies found to support your conclusions, from references 8 ,11-13

Line 166.  Since “the accurate and timely determination of meth resistance is key…”  it seems like treatment is important.  Are these shelter dogs quickly and effectively treated with one of the antibiotics which show sensitivity to Staph?

Line 167  Is this discussing human spread?  Sounds like dogs are getting it from people

Line 170  “…risk factor…”  What are other risk factors?  Are humans living together in over crowded spaces a risk factor ? What about hospitalization and medical procedures?  How do these compare with the dog risk factor?  Previous papers show human carriage rate can range between 20-60%  Although this is addressed in 184, it should be included earlier and doesn’t need to be it’s own paragraph

Not sure if it would be possible or could fit well into Table 5 or SI 1 but It would be valuable to know which antibiotics were most valuable in preventing resistance eg antibiotics of last resort . Is Vancomycin still one of those antibiotics?

Table 1 Females > males.  Any lactating?  Pregnant?

Table 2 needs to be broken down into stray and owned animals

Table 4.  Did any dog have multiple lesions eg both puritis and erythema?  Was there a  statistically significant different between stray and owned dogs? Female and Male dogs?

Table 5 Possibly add significance of antibiotic eg antibiotic of last resort in numerical order

Conclusion

  Are people spreading it to dogs?  It is stated that MRSA is of high concern but 5/43 were positive for MRSA.      Not sure the use of “extremely alarming matter” is correct since human to human transmission is much more of a concern.   How likely is S aureus to be transmitted via manure or poor waste management practices?

M and M

Why was June-Oct picked for sampling? Why such a short time frame? What is  the weather like during that time?  Would one expect the weather during this time of year to exacerbate skin conditions?

Why were the stray dogs hospitalized?  Were they stray pet dogs, community owned dogs?   Were any feral eg unsocialized and aggressive when humans attempted to work with them?

Would suggest reorganizing this section.  There are 5 paragraphs in this section and would suggest beginning with the paragraph at line 226, followed by 224, 233, 217, then 238

Line 233   This study recruited from the animal shelters, is that correct?  They were not collected off the streets for this study-correct?

Line 238.   What is a “dermatological stray dog”?

Statistics should be included in this paper

Author Response

We want to thank them for their valuable comments and suggestions which have been of help to improve the quality of the manuscript. Below, you can find the authors' answers regarding their requirements and observations.

Reviewer 2 Report

  • Discussion is focused only on few MRSA isolates of the study. MRSP should also be discussed, as well as their biological significance as zoonotic agents. 
  • Discussion section should be extended, including the incidence of MRSP in several European countries and USA, the misidentification of S. pseudintermedius as S. aureus and appropriate criteria for avoiding this to happen.
  • What strains of MRSP have been isolated ? (ST71, ST68, other?)   
  • There are no dermatological signs in the dog. Instead, there are skin lesions
  • The terms "dermatological dogs" and "non-dermatological dogs" are not appropriate. They must be replaced. There are dogs with or without skin lesions, respectively
  • Materials and Methods section: The methodology applied should be referred in brief followed by appropriate citations.
  • Zoonotic implications for isolates should be further stressed, along with control options for colonized healthy dogs.

Author Response

Many thanks for your valuable comments and suggestions which have been of help to improve the quality of the manuscript.

Reviewer 3 Report

The reviewed manuscript is aimed on the interesting microbiological issue of Staphylococcus sp. isolated from stray dogs skin. As well-known this bacteria species is still significant problem both in human and animal medicine. The presentation of results is sufficient and significant for further microbiological and epidemiological studies (antibiotics resistances and zoonoses). Even that it is well written I have some remarks:

  1. It would be good to add some figures of isolated bacteria colonies and some basic microbiological staining (i.e. Gramm)
  2. It would be good to add some figure of pathological changes of dog’s skin observed in exanimated animals.

I recommend to accept corrected manuscript for printing after minor revision.

Author Response

We want to thank you for your suggestions which have been of help to improve the quality of the manuscript.

Reviewer 4 Report

There are no conclusions in the abstract.

The study was not conduced on stray dog per se, but on shelter dogs in Timisoara. I suggest to change the title on “Carriage of multi-resistant Staphylococci in shelter dogs in Timisoara”

 It is not clear how long these dogs were in the shelter and how they were kept, individually or in groups. Please clarify this.

Lines 61-62; 144 – please delete this sentences.

Line 120 - The majority of the 43 studied Staphylococcus isolates… . How many, exactly by number and percentage?

There are elements of discussion in the Results (lines 92-94; 113-116; 152-154), please move them to the Discussion.

The discussion consists largely of one-sentence paragraphs. Please combine them into larger one

The possible sources of antimicrobial resistance of Staphylococcus spp.in shelter dogs should be discussed. Have these dogs been treated with antibiotics?

Author Response

We want to thank you for pertinent comments and suggestions which have been of help to improve the quality of the manuscript.

Round 2

Reviewer 1 Report

This manuscript has been improved tremendously.  Although not required, part (lines 164-182) of the discussion still reads like bullet points of a meeting poster and has a disjointed feel.  I would suggest trying to combine to improve the flow. After a minor review for spelling and grammar it is appropriate to be published.

Author Response

Again, we wish to show our gratitude for the valuable time spent with our manuscript. improve its quality, and for all the priceless recommendations possed.
